# Comment on Choi, Y.-J., et al. Cellular Phone Use and Risk of Tumors: Systematic Review and Meta-Analysis. *Int. J. Environ. Res. Public Health* 2020, *17*, 8079

**DOI:** 10.3390/ijerph18063125

**Published:** 2021-03-18

**Authors:** Frank de Vocht, Martin Röösli

**Affiliations:** 1Population Health Sciences, Bristol Medical School, University of Bristol, Bristol BS8 2PS, UK; 2Department Epidemiology and Public Health, Swiss Tropical and Public Health Institute, 4002 Basel, Switzerland; martin.roosli@swisstph.ch; 3University of Basel, 4003 Basel, Switzerland

We welcome the updated systematic review and meta-analysis of case-control studies of mobile phone use and cancer by Choi et al., which was recently published in this journal [1]. Given the uncertainties that remain to surround the issue of radiofrequency radiation exposure and cancer risk, regular synthesis of available epidemiological evidence continues to be important, and the synthesis published by Choi et al. provides a timely update. However, Choi et al. have made several peculiar decisions in their synthesis which result in difficulties in the inferences that can be made, and which deserve further discussion.

Firstly, the main meta-analysis shown in Figure 2 in [1] combined case-control studies for different benign and malign tumors, including those of the head, but also non-Hodgkin’s lymphoma and leukaemia, and provides one meta-analytic summary of these. It is not common practice to combine different outcomes with different aetiologies in one meta-analytic summary [2,3] and, given the substantial heterogeneity observed, it is highly questionable if the common risk estimate for diseases with different aetiologies that Choi et al. try to combine in their meta-analysis does exist (note that the arbitrary set of outcomes used by Choi et al. is not the same as ‘all cancers’, which is an aggregate outcome used in meta-analyses). It would be more appropriate to conduct separate meta-analyses by type of tumor, and Choi et al. have indeed done these as well. These results are provided in the Online Supplement (Table S3) and do not provide summary evidence of excess tumor risk for any particular individual tumour types. 

Choi et al. further presented subgroup analyses of studies conducted by Hardell et al., studies by the INTERPHONE consortium, and a group of miscellaneous case-control studies. They identify interesting differences between those three subgroups, and conduct further analyses to explore possible reasons for the observed differences. Interestingly, Choi et al. fail to notice the most obvious conclusion from these subgroup analyses, in that both the INTERPHONE-related studies and miscellaneous studies are largely in agreement and do not point to an excess cancer risk from mobile phone use. Evidence of large excess cancer risks are almost exclusively based on the studies by the Hardell group; as already described in earlier meta-analyses [4,5]. In fact, relative excess risks of 90% (30–170%) and 70% (4–180%) reported by the Hardell group (Table 1 and Figure 2) associated with any mobile phone use are implausible high, and do not triangulate [6] with evidence from other epidemiological sources, such as prospective cohort studies [7,8] and incidence trends [9]. Incidence trend analyses are generally considered a weak study design but in this specific case of a clear change of exposure of virtually the whole population, limited confounding factors that may change over time and reliable cancer registries, incidence trends are important for evidence evaluation and plausibility considerations. 

Even when exposure-response associations are observed (Table 3), and the INTERPHONE studies and miscellaneous studies provide relative consistent estimates (Odds Ratios of 1.25 (0.96–1.62) and 1.73 (0.66–4.48), respectively) of some excess risk associated with a, arbitrary, cumulative call time of at least 1000 h, the evidence from the Hardell studies similarly provides an implausibly high Odds Ratio of 3.65 (1.69–7.85); out of line with all evidence from other sources. The INTERPHONE team have spent considerable efforts trying to evaluate whether observed increased and decreased risks could be the result of recall and selection bias [10,11,12,13,14] and a recent study found some indication for reverse causality as an explanation for seemingly protective effects from mobile phone use [15]. It is therefore surprising that Choi et al. have not similarly discussed the likelihood of bias away from the Null in the Hardell studies. Disregarding the implausible risk reported by the Hardell group, a summary risk point estimate based on all other case-control studies for 1000+ cumulative hours of use would be in the order of 1.30–1.50, which triangulates much better with other lines of research. 

Choi et al. argue that a plausible explanation for the observed differences could be that the Hardell studies are of better quality than those in the other two groups, based on individual appraisal of each study using the Newcastle-Ottawa Scale and National Heart, Lung, and Blood Institute quality assessment tool of case-control studies (Tables S1 and S2). The differences in rating within and between the three groups of case-control studies are minimal, but Choi et al. rated the methodological quality Hardell studies a little higher quality mainly because they had very high response rates and because they mostly classified as having excellent, blinded, assessment of exposure compared to the INTERPHONE and miscellaneous studies. This seems to be an error or misunderstanding in the use of these criteria. First, achievement of a high participation rate is an asset in an epidemiological study. However, to achieve a participation rate of over 80% in a population based case-control study in Western Countries, as reported in the Hardell papers, is highly unusual nowadays. Regardless, one would expect that in a study with such high participation rates, the proportion of mobile phones users in controls should closely match the official subscriber statistics, which was not the case for the Hardell studies [5]. Thus, serious concerns remain about how these high participation rates have been achieved or calculated.

Secondly, the blinding concept as rated by Choi et al. is inappropriate. Exposure assessment in the INTERPHONE studies was conducted by trained interviewers, which have been susceptible to interviewer bias because they could indeed probably not be blinded to case-control status [16]. However, it is highly unlikely this would have resulted in higher bias compared to the Hardell studies, in which exposure assessment was based on questionnaire-based self-reporting, by cases and controls, of mobile phone use who, by definition, are not blinded to their disease status. Methodological work suggests that both face-to-face interviews and self-administered questionnaires are susceptible to various ‘mode of administration’ biases, but that exposure assessment based on self-administered questionnaires are generally more susceptible to recall bias [16]. As such, the methodology of the Hardell studies should have been classified as being of comparable quality to the other case-control studies in this review, at most.

Choi et al. further looked at source of funding as a possible explanation for observed differences, but provided erroneous funding information. Only the Hardell studies received direct funding from interest groups such as the telecom industry [17,18] and pressure groups [19], while Hardell has also acted as an expert witness on behalf of the plaintiff in hearings involving mobile phone use and brain tumour risk in the past, presumably reimbursed [20], but this was not reported by Choi et al. In contrast, INTERPHONE studies’ industry funding was through a well-established firewall model to avoid influence of the funders on the researchers. There is empirical evidence from human experimental studies that such a funding structure has not resulted in biased study results but in higher study quality, whereas direct funding by interest groups may produce biased results [21,22]. Further, the three study groups only contribute to either ‘funded by industry’ or not (according to Choi et al.), which makes this analysis non-informative. 

Given that observational epidemiological studies are susceptible to various biases, which can result in under as well as over-reporting of true effects, rigorous evaluation is needed to understand why the studies by the Hardell group provide different results than the majority of other case-control studies and with other groups of epidemiological literature. In the absence of direct evidence for any causes of these differences, triangulation of epidemiological studies susceptible to different types of biases [16], such as case-control studies, cohort studies, and ecological studies of cancer incidence, as well as with evidence from animal and laboratory studies is warranted. Although some uncertainties remain, most notably for highest exposed users and for new GHz frequencies used in 5G, we can be reasonably sure that the evidence has converged to somewhere in the range of an absence of excess risk to a moderate excess risk for a subgroup of people with highest exposure. Important, over time, the evidence had reduced the uncertainty regarding the cancer risk of mobile phone use.

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
