# Peer review of "Comment on Choi, Y.-J., et al. Cellular Phone Use and Risk of Tumors: Systematic Review and Meta-Analysis. Int. J. Environ. Res. Public Health 2020, 17, 8079"

_ijerph, 2021, doi:10.3390/ijerph18063125_

Round 1
Reviewer 1 Report
The claim by Vocht and Röösli that you cannot combine studies aimed at different tumor types is not tenable. A large part of the epidemiology literature studies a relation between one agent and one tumor, mostly because of methodological issues. It is work-intensive to obtain and purify reliable hygiene and tumor data, and focusing on a fraction of the problem decreases costs, while increasing data reliability, and the chance of coming up with useable results. Toxicology has a long history of studying chemicals, and because a chemical is often absorbed by a specific route of entry, metabolized, and compartmentalized often according to the Pow of the chemical or its metabolites, it is not uncommon to find evidence that one agent preferentially targets one or a few organs. In the case of electromagnetic radiation, which penetrates the body more or less uniformly, without the limitations of chemicals in terms of route of entry and compartmentalization, it is perfectly justifiable to assess all targets simultaneously, as was done by Choi. If you followed Vocht and Röösli's specificity principle, and looked at each type of tumor and each frequency specifically, you would thin out the data to such an extent that no significant conclusion could ever be reached. So, if fact, Choi's approach is the only feasible one. Using both benign and malignant tumors is appropriate, because no agent was ever proven to produce benign tumors specifically. Choi selects cellular phone use as an indicator of nnEMF, that is, non-natural ElectroMagnetic Field exposures. In view of the complexity of the present EMF environment, it is not surprising that effects require skill to discover, because it seems that deeply unexposed subjects are almost impossible to find, or have other lifestyle peculiarities that make them poor controls. As pointed out by Choi, some of our largest public health foes (smoking) have multiple targets. I could teach a complete Toxicology course illustrating, in organ after organ, the toxic effects of a single agent, lead. >From the Vocht and Röösli discussion of the Hardell studies, I will retain that because Hardell received funding from the telecom industry, he actually downplayed cellular phone risks in his work. And pushing the distinction between direct (open) and firewall funding (a setup to ostensibly separate) does not strengthen Vocht and Röösli’s argument, but makes it more suspect. Henry Lai and Lennart Hardell have already demonstrated the influence of industry in no equivocal terms (1, 2). When the circumstances of epidemiology are so difficult, it is time to seek guidance from animal experiments, which can be more tightly controlled, and also on basic physiology, which can be even more controlled. As Choi points out, the verdict from rats and mice has been very strongly in favor of drastically reducing EMR exposures (reference 33 and 34 of Reply), and this verdict has been continuously maintained over decades (3, 4, 5). My own laboratory experiments (6, 7, 8) and those of countless others prove that the idea of viewing impacts of EMR a heat only is untenable, and completely out of line with the traditions of risk assessment. So the question arises, why does EMR get such special treatment? When discussions become mired into very small details, possibly to create the illusion of a controversy, it is sometimes useful to take a step back, and look at the historical timeline of EMR exposures. 1946 Commercialization of microwave ovens, emitting a completely new and artificial agent in the environment. 1966 High limits for human exposures are adopted by the military, on the basis of short-term experiments, while admitting to some poorly understood outlying risks. This, at a time when human exposures were small, and subjects exposed were few. 1983 Start of huge exposure rises to humans due to proliferation of cell phones. 1977 Creation and promotion of a selected group of people (ICNIRP) to maintain the exposure limits at military levels, reflecting the need of industry to stabilize standards and secure investments in wireless. The last phrase in Vocht and Röösli's letter is quizzical: "Important, over time, the evidence had reduced the uncertainty regarding the cancer risk of mobile phone use." What does this mean? Has the evidence of harm from EMR become stronger, or weaker? If they Vocht and Röösli believe the evidence of harm is stronger, congratulations to them. If they believe the evidence of harm is weaker, they have not read the National Toxicology Program or Ramazzini Institute studies (reference 33 and 34 of Reply). In such a case, their views are prisoners of the distant past. I would rather eat Choi's fruit salad than Vocht and Roosli's moldy dish. 1. https://bioinitiative.org/ Bioinitiative report 2012, page 16. https://www.seattlemag.com/article/uw-scientist-henry-lai-makes-waves-cell-phone-industry 2. Health risks from radiofrequency radiation, including 5G, should be assessed by experts with no conflicts of interest. LENNART HARDELL and MICHAEL CARLBERG. Oncol Lett. 2020 Oct; 20(4): 15. Published online 2020 Jul 15. doi: 10.3892/ol.2020.11876 3. Chou 1992 CK et al. Long-term, low-level microwave irradiation of rats. Bioelectromagnetics. 1992;13(6):469-96. 4. Lerchl 2015 et al.Tumor promotion by exposure to radiofrequency electromagnetic fields below exposure limits for humans. Biochemical and Biophysical Research Communications 459 (2015) 585- 590. 5. Repacholi 1997 MH et al. Lymphomas in Emu-Pim 1 Transgenic Mice Exposed to Pulsed 900 MHz EM Field. Rad Res 147:631-640. 6. Li 2012 Y, Héroux P, Kyrychenko I. Metabolic Restriction of Cancer Cells in vitro causes Karyotype Contraction - an indicator of Cancer Promotion? Tumor Biology 2012; 33(1):195-205 DOI:10.1007/s13277-011-0262-6. 7. Li 2013 Li and Paul Héroux. Extra-Low-Frequency Magnetic Fields alter Cancer Cells through Metabolic Restriction, Electromagnetic Biology and Medicine 33(4):264-75. DOI:10.3109/15368378.2013.817334, 2013. http://www.tandfonline.com/doi/full/10.3109/15368378.2013.817334. 8. Li 2019 Li and Paul Héroux. Magnetic Fields Trump Oxygen in Controlling the Death of Erythro-Leukemia Cells, Appl. Sci. 2019, Volume 9, Issue 24, 5318. https://www.mdpi.com/2076-3417/9/24/5318/pdfReviewer 2 Report
de Vocht, Röösli
This seems to be a scientifically unfounded attack on the Hardell group studies on this issue. Thus it is not suitable to be published in a scientific journal but should be rejected. There are many statements that are not qualified according to published studies. Inclusion of references is selective without a comprehensive review of the consistent pattern of increased risk.
The authors claim that excess cancer risks “associated with any mobile phone use are implausible high”. This is however not correct. A meta-analysis published in 2018 shows ORs in the Hardell group that are even lower than by Coureau et al (2014) and well within CIs for the Interphone studies. This is neglected by Röösli and de Vocht.
(Belpomme D, Hardell L, Belyaev I, Ernesto Burgio E and Carpenter DO: Thermal and non-thermal health effects of non-ionizing radiation: an international perspective. Env Poll 242: 643-658, 2018)
”such as prospective cohort studies [6, 7]”
The Benson et al was not a prospective study on mobile phone use. Exposure was assessed at one point and not part of the initial study. The Danish cohort study [7] was by IARC in 2011 evaluated to be uninformative due to serious misclassification of exposure. Röösli was part of that decision group. Thus it should not be included as scientific evidence.
Regarding incidence trends by Karipidie et al [7] it is important to notice, but neglected Rööslii and de Vocht that only brain tumour data for ages 20-59 were reported. This represents about 39% of Australian brain tumours, 6% of cases are under age 20 and 55% are aged over 59. Interphone and other studies have reported that long-term use and latency are important factors, it is inappropriate to exclude the group of people who will generally have used their phones for the highest number of years, and who are also the age-group who already had the highest incidence of brain tumours.
In addition, Karipidis et al age-standardised their data to the World Health Organisation's (WHO) world standard population, which in no way represents the modern Australian population age-spectrum. It over-weights young ages and significantly under-weights ages over 45. If age-standardised (rather than age-specific) data is to be used when investigating modern trends in a current population, then the standard population spectrum used should reasonably match the current actual population - especially with the current rapid increase in elderly people. See (Philips A. Significant flaws and unjustifiable conclusions BMJ Open Jan 2019)
In contrast to that the authors have excluded reference to incidence data in England by Philips et al (2018) and Sweden by Hardell et al (2015).
It is unclear why the results in the Hardell group studies are ‘implausible’. As shown by Belpomme et al (2018) they are in agreement with other results. Moreover the authors have omitted further results with increased risks based on the Interphone study results in:
Cardis E, Armstrong BK, Bowman JD, Giles GG, Hours M, Krewski D, McBride M, Parent ME, Sadetzki S, Woodward A, et al: Risk of brain tumours in relation to estimated RF dose from mobile phones: results from five Interphone countries. Occup Environ Med 68: 631–640, 2011.
Grell K, Frederiksen K, Schüz J, Cardis E, Armstrong B, Siemiatycki J, Krewski DR, McBride ML, Johansen C, Auvinen A, et al: The Intracranial Distribution of Gliomas in Relation to Exposure From Mobile Phones: Analyses From the INTERPHONE Study. Am J Epidemiol 184: 818–828, 2016.
Momoli F, Siemiatycki J, McBride ML, Parent MÉ, Richardson L, Bedard D, Platt R, Vrijheid M, Cardis E and Krewski D: Probabilistic Multiple-Bias Modeling Applied to the Canadian Data From the Interphone Study of Mobile Phone Use and Risk of Glioma, Meningioma, Acoustic Neuroma, and Parotid Gland Tumors. Am J Epidemiol 186: 885–893, 2017.
However, to achieve a participation rate of over 80% in a population based case-control study in Western Countries, as reported in the Hardell papers, is highly unusual nowadays.
Recruitment of cases and controls in the Hardell group studies were closed in 2009. Use of mobile phones was assessed from the 1980s until not later than 2008. Thus this does not represent use ‘nowadays’. The response rate in e.g. parts of Interphone in Sweden were even higher than in the Hardell group, see (Hardell et al Occup Environ Med. 2005)
However, it is highly unlikely this would have resulted in higher bias compared to the Hardell studies, in which exposure assessment was based on questionnaire-based self-reporting, by cases and controls, of mobile phone use who, by definition, are not blinded to their disease status.
The authors make a mix of observational bias and recall bias. As to observational bias the Hardell group assessed exposure by self-administered questionnaires sent to cases and controls and supplemented over the phone if necessary. No data on case or control status were shown for the interviewers. In Interphone personal, even bedside interviews, were used which must have been a stressful situation for the patient compared to fill in the questionnaire in a more relaxed situation at home. Interphone has not displayed how these personal interviews were performed regarding controls in large geographic areas. Did the interviewers make long trips or were controls selected nearby the hospitals.
Regarding recall bias the Hardell group included glioma, meningioma and acoustic neuroma in the same studies. Different results were obtained for different tumour types in the same study. Using the meningioma cases as ‘controls’ to the glioma and acoustic neuroma cases still yielded statistically significant increased risks for these tumour types. These results have been published but are omitted by de Vocht and Röösli, see (Hardell et al Pathophysiology 2015, and Hardell et al Int J Oncol2013).
Regarding direct funding – according to the studies that was obtained for one part of the studies. The results were similar in all studies with increased risks. Thus no bias existed.
..as well as with evidence from animal and laboratory studies is warranted..
The authors have excluded the animal study results in the NTP studies and Ramzzini Institute study. These are published as well as evidence on oxidative stress and DNA damage of importance in carcinogenesis.
Finally Röösli does not report his membership in ICNIRP as a potential conflict of interest.
It should be noted that the Ethical Board at the Karolinska Institute in Stockholm, Sweden concluded already in 2008 that being a member of ICNIRP may be a conflict of interest that should be stated officially whenever a member from ICNIRP makes opinions on health risks from EMF on behalf of another organization, as in this case (Karolinska Institute Diary Number 3753-2008-609).
Reply by Myung et al
This is a well-balanced reply. The statements are adequate based on the scientific literature on this issue.
There are no further comments on the reply.